# Hygrothermal Damage Monitoring of Composite Adhesive Joint Using the Full Spectral Response of Fiber Bragg Grating Sensors

**DOI:** 10.3390/polym14030368

**Published:** 2022-01-18

**Authors:** Chow-Shing Shin, Tzu-Chieh Lin

**Affiliations:** Department of Mechanical Engineering, National Taiwan University, No. 1, Sec. 4, Roosevelt Road, Taipei 10617, Taiwan; R04522534@ntu.edu.tw

**Keywords:** adhesive joint, hygrothermal damage, fiber Bragg grating, tensile failure, fatigue failure, full spectral response

## Abstract

Adhesive joints in composite structures are subject to degradation by elevated temperature and moisture. Moisture absorption leads to swelling, plasticization, weakening of the interface, interfacial defects/cracking and reduction in strength. Moisture and material degradation before the formation of defects are not readily revealed by conventional non-destructive examination techniques. Embedded fiber Bragg grating (FBG) sensors can reflect the swelling strain in adhesive joints and offer an economical alternative for on-line monitoring of moisture absorption under hygrothermal aging. Most of the available works relied on the peak shifting phenomenon for sensing. Degradation of adhesive and interfacial defects will lead to non-uniform strain that may chirp the FBG spectrum, causing complications in the peak shifting measurement. It is reasoned that the full spectral responses may be more revealing regarding the joint’s integrity. Studies on this aspect are still lacking. In this work, single-lap joint composite specimens with embedded FBGs are soaked in 60 °C water for 30 days. Spectrum evolution during this period and subsequent tensile and fatigue failure has been studied to shed some light on the possible use of the full spectral response to monitor the development of hygrothermal degradation.

## 1. Introduction

The advantageous specific stiffness and specific strength and the excellent corrosion resistance of fiber reinforced composites lead to their increasing replacement of metallic materials in adverse environments. Examples of these include applications in the petroleum and gas extraction sector [1,2], tidal turbine blades [3], wind turbine blades [4], air [5,6] and land transport vehicles [7] as well as cooling towers [8]. Adhesive bondings are widely used to join components in these structures in preference to bolting and riveting as they keep the structure surface smooth, distribute and reduce joining stresses and help to avoid fiber discontinuity [9,10]. However, one disadvantage of adhesive joints is the difficulty in examining its integrity. The elevated temperature and humidity often encountered in adverse environments are well known to degrade polymers through a hygrothermal aging process [11,12,13,14] and poses a major concern on the joint durability. The coupled effect of temperature and moisture affects the adhesive joint more than the composite [15]. Service loading conditions such as impact, occasional overload and fluctuating loading will further aggravate the damages. If such degradation went undetected, serious structural failures and catastrophic outcome might follow.

Moisture diffused into polymers will be absorbed as free water and bound water [16,17,18]. The former occupies the micro-cavities of the polymer network while the latter forms hydrogen bonds with polar segments of the polymer molecules. Bound water will cause swelling of the polymers, but free water will not. Besides swelling, the absorbed water will cause plasticization, decrease in the glass transition temperature and reduction in their mechanical strength [16,17,18,19]. Eventually, it will also lead to weakening of the interface and interfacial defects/cracking [11,12,13,19,20,21,22]. Elevated temperature will accelerate moisture diffusion [20,23,24] and aggravate degradation on the one hand. On the other hand, in thermoset adhesive with incomplete cross-linking after cure, elevated temperature may enhance post-curing which increases cross-linking and strength [13,24,25,26] as well as reduces volume [14,21].

A number of non-destructive examination techniques have been attempted to reveal defects in adhesive joints. These include traditional ultrasonic techniques [27,28,29], guided wave [30,31], acoustic microscopy [32,33], electromagnetic acoustic transducer [34], electromechanical impedance spectroscopy [35,36,37], thermography [29,38,39] and shearography [28,40,41]. Davis and McGregor [42] pointed out that most non-destructive inspection for defects techniques are mainly useful at the fabrication stage and at the late stage of failure when explicit defects have formed, but are ineffective during the joint material degradation stage. Additionally, the presence of moisture cannot be easily detected by conventional non-destructive techniques [43].

Moisture content was sometimes monitored as an indicator of the degree of damage. Experimentally, a commonly used method is gravimetric analysis [13,43,44,45]. However, gravimetric weighing of an engineering structure is impracticable. Furthermore, gravimetric analysis is not so straightforward for adhesive joints that involves composite materials, as both the adhesive and the composite matrix will absorb moisture. Local moisture content can be monitored by embedded interdigital electrode sensor that made use of impedance changes [46] and optical fiber evanescent sensor that depended on light energy loss with surrounding refractive index changes [47,48]. Computational methods have been developed to predict the diffusion of moisture and the swelling strain under known boundary conditions [13,43,44,45,46,47,48,49]. The calculated moisture contents agreed well with the gravimetrically measured ones [13,43,44]. In practical applications, the environmental humidity and temperature are varying from time to time and often not precisely known as that in experimental investigations. This makes the application of computational technique difficult. Moreover, as pointed out above, a number of interacting damaging mechanisms are operating and so the moisture content and the properties degradation is not related in a straightforward manner.

Instead of monitoring moisture content or directly looking for defects, there are techniques suitable for real time monitoring of the integrity degradation of adhesive joints. These include strain/stiffness monitoring using back face strain gages [49,50,51], resistance monitoring of adhesive joints that were made conductive by adding carbon nanotubes [52,53] and optical fiber sensors signal surveillance [43,44,54,55,56,57,58,59,60,61,62,63,64,65,66,67,68,69,70,71,72,73]. Strain gages can only be applied to the outer surface, they will disrupt an otherwise smooth surface and are susceptible to environmental degradation. Additionally, they may not possess sufficient fatigue life for long-term monitoring [74]. Resistance method is economical to deploy to large adhesive joints, but it is not easy to locate the damage sites. Its applicability to adherends with high resistivity may be limited. Optical fibers are known to have excellent fatigue endurance. They can be embedded inside the bond to leave the external surface smooth, which is important for aerodynamic structures. They are relatively free from environmental attack and have been used for general structural health monitoring [54,55,56]. There are different kinds of optical fiber sensors. In adhesive joints, both the distributed sensing [57,58,59] and the discrete fiber Bragg grating (FBG) sensors have been used [30,60,61,62,63,64,65,66,67,68,69,70,71,72,73]. The distributed sensors measured strain along the fiber and is equivalent to a train of strain gages with the best available spatial resolution going down to the millimeter range. Discrete FBG sensors encode measurand information in the wavelength of a characteristic spectrum. The uniform period FBG and the chirp FBG have both been used. The chirp FBG has a programmed distribution of grating periods that offer a certain degree of spatial resolution. It has been shown that chirp FBGs embedded in adhesive joints can detect the occurrence and location of artificially induced [67] or naturally initiated disbonds [68,69].

The uniform period FBGs were often treated as embedded strain gages to monitor the deformation of adhesive joints under mechanical load [60,61,62,63,64,65,66] and hygrothermal swelling [43,44,72,73]. When used in this way, the peak wavelengths were usually logged and converted to strain. However, automatic peak wavelength loggers will normally lock on to the peak with the strongest intensity. FBGs embedded in adhesive joints sometimes showed spectrum splitting on adhesive curing [21,43,72,75]. Moreover, absorption of moisture may cause heavy chirping [30]. This makes the identification of a single representative wavelength from the spectrum impracticable. The employment of a single-peak wavelength obviously cannot reflect the actual strain status of the bond.

The use of FBGs for hygrothermal aging monitoring has a number of distinct advantages over conventional sensors. These include a much better compatibility with the host materials and will not behave as defects when embedded, the ease of multiplexing, as well as immunity from electromagnetic interference and environmental attack. Moreover, the grating period that responds to the surrounding strain is in the order of micrometer. When embedded, its close proximity to the degraded materials/defects together with its small responding gage lengths make FBGs very suitable for reflecting small local perturbation of strain caused by material changes and degradation. Instead of the normally employed single-peak wavelength that only indicates the swelling strain, it is postulated the full spectral response of FBGs will be much more capable in revealing the onset of hygrothermal degradation as well as the development of subsequent aging damages. The clearer picture about the degradation status so obtained can help to avoid uneconomical premature retirement of components and precarious use of heavily degraded structures.

At present, adhesive joint integrity monitoring making use of the full FBG spectral responses are extremely limited and mainly used on mechanical loading [30,70,71]. Webb et al. [70,71] applied a dynamic full-spectrum interrogator to a single-peak FBG sensor embedded in the adhesive of a single-lap joint under cyclic loading. The dynamic interrogator logged the full spectral response via an intensity modulated set-up [74] and extracted the peak wavelength information. This, in effect, used the FBG as a strain gage with very high frequency response rather than making use of the full information in the whole spectrum. Karpenko et al. [30] attempted to use the information of the full chirped spectrum, but their adhesive joint had not yet been loaded to the point of damage initiation. The capability of the FBG spectrum to reveal hygrothermal aging damage is still not clearly known.

In view of the limitations of the available results on these aspects, hygrothermal aging followed by tensile and fatigue tests to failure will be carried out on adhesively bonded single-lap joint specimens. The capability of the full spectral responses of single-peak FBGs to monitor hygrothermal aging, to detect the onset and follow the development of damages incurred during the aging stage and the subsequent mechanical loading stage, will be investigated.

## 2. Materials and Methods

### 2.1. Fiber Bragg Grating Sensor and Its Basic Properties

A fiber Bragg Grating (FBG) is a certain section on an optical fiber with a periodic variation of refractive index. With a uniform period Λ, the grating will reflect a characteristic single narrow-peak spectrum with wavelength λ from an incident broadband light [76]:Λ = 2nΛ (1)
where n is the effective refractive index of the fiber core. When a freely standing FBG is subjected to a uniform longitudinal stress/strain, Λ will change due to the resulting strain and *n* will change by the photoelastic effect [76]. This causes a change in λ and the reflected spectrum will shift as a whole, as shown schematically in Figure 1a. Transverse stresses acting on the FBG will bring about birefringence effect, resulting in a splitting of the spectrum peak (Figure 1b) [77]. If the stress/strain is varying along the length of the FBG, different λ’s will satisfy the condition of Equation (1) on a different sections of the grating. As a result, the grating acts as a series of FBGs with different peak wavelength λ’s, thus the spectrum broadens or chirps, as shown schematically in Figure 1c. The shape of the chirped spectrum will be governed by the pattern of stress distribution.

When the freely standing FBG is subjected to a temperature change instead of stress, Λ will be changed by thermal expansion/contraction and *n* will be changed through the thermo-optics effect [76]. These will also cause a change in λ and a shift of the whole reflected spectrum such as that shown in Figure 1a. Typically for an FBG with λ = 1550 nm, a tensile strain of 1 με or 1 °C rise shift the spectrum by ~1 pm or ~10 pm, respectively, towards the longer wavelength. The above effects will superimpose if both temperature change and mechanical stress are applied to the FBG. When a compressive strain or a drop of temperature occur, the spectrum shifts in the opposite direction towards the shorter wavelengths.

The above discussion is for a freely standing FBG. For an FBG embedded in an adhesive joint, more complications arise. Firstly, the curing process is a chemical reaction that often involves temperature changes and the progressive solidification involves volume contraction. The rate of cure depends on a number of factors such as temperature, humidity and stoichiometric ratio [78,79]. Local variations of these factors invariably exist and differential curing rate at different vicinities of the FBG and at different locations along the FBG will lead to a varying residual stress distribution acting on the grating, modifying the reflected spectrum. Existence of local variation of residual stress may best be illustrated in an observation that spectrum chirping/splitting occurred in a 10 mm-long FBG but not in a 1 mm FBG while both were embedded in parallel in the same joint [72]. In fact, peak splitting and spectrum chirping are commonly observed when uniform period FBGs are embedded in adhesive joints [21,43,72,75].

External load and temperature variation on the joint specimen act not only on the optical fiber but also on its surrounding adhesive, leading to a redistribution of the residual stress on the FBG. The resultant effect on the change of the FBG spectrum will therefore come from three contributions: (1) stress distribution in the joint caused by the applied load; (2) temperature change of the FBG; (3) residual stress distribution along the FBG. When damages occur in the joint, they will act as local stress raisers. These will affect the load-induced stress distribution as well as causing further redistribution of the residual stress. The resulting FBG spectrum will therefore be affected by a superpositioning of the above effects.

In this work, single-peak FBGs were used and were fabricated in a Ge-B co-doped single-mode optical fiber by side writing using a phase mask [80]. The sensing length of the FBGs was about 10 mm. The reflectivity of the as produced FBG was ~99%. The reflected spectra from the FBGs were interrogated using an optical spectrum analyzer (MS9710C, Anritsu, Kanagawa, Japan). Mechanical testings were carried out in an air-conditioned room with thermostat control set to ±1 °C and the FBGs were embedded in poor thermal conductors of polymeric adhesive sandwiched between composite laminates so that it is relatively insensitive to outside temperature changes. Small ambient temperature fluctuations will have negligible effect on the measured spectra during mechanical testing.

### 2.2. Single Lap Joint Specimens

101.6 mm × 25.4 mm strips were cut from a 220 mm × 220 mm Graphite-epoxy composite laminate consisting of 10 unidirectional plies. Each two of these strips were glued together with Loctite structural epoxy adhesive (E-30CL, Henkel Taiwan Ltd., New Taipei City, Taiwan) to form single-lap joint specimens. The fiber direction in the composite is along the loading axis. The areas to be joined were sanded and masking tape was applied to the immediate vicinity beyond the boundary of the joint area. The purpose of the tape is to prevent excess glue resulting in additional but unpredictable adhesion between the two parts. Excess glue was difficult to clean off especially with the optical fibers in place. Three optical fibers with FBGs were embedded in the joint, as sensors as well as spacers. The bond line was approximately as thick as the diameter of the optical fiber, i.e., 125 μm. A section of the same composite strip was also glued to each end of the specimen to ensure the loading axis to pass through the center of the adhesive layer. Detailed dimensions and layout of the specimens are shown in Figure 2.

A batch of seven single-lap joint specimens can be made from a 220 mm × 220 mm composite laminate. Preliminary tests showed that their tensile strengths are affected by the environmental conditions, such as temperature and humidity during the joining operation. Different specimen batches had average batch strengths from 8.26 to 10 kN. However, within the same batch, the worst-case standard deviation of tensile strength was within 4.1%. Thus, for each batch of seven specimens, two randomly sampled specimens were tested under monotonic loading to obtain the average batch tensile strength while the remaining specimens were used for hygrothermal treatment and various testing and measurements.

### 2.3. Hygrothermal Treatment

Single-lap joint specimens were incubated in 60 °C water for 30 days. A 6.7 L recirculating water bath (B401, Firstek Scientific, Taipei, Taiwan) was employed for this treatment. The temperature of the water was controlled to within ±0.1 °C. The FBG spectra during the heating up, soaking and cooling down processes were recorded. After soaking for 30 days, the specimens were taken out of the bath and wiped dry. Most of the specimens were subjected to mechanical testing immediately in this as-soaked condition. The duration between leaving the water bath and starting tensile and fatigue tests was within 1 h. A limited number of specimens were put into a desiccator and dried for 10 days before undergoing mechanical testing.

### 2.4. Mechanical Testing

Specimens were subjected to tensile or cyclic loading on a servo-hydraulic testing machine (810 Materials Testing System, MTS Systems, Eden Prairie, MN, USA).

During the tensile test, loading was periodically interrupted to allow the reflected light spectra from the FBGs to be recorded at the instantaneous loading. The specimen was then unloaded to allow reflected spectra to be measured at 0 N. This loading–unloading cycle was repeated with progressively higher loading until specimen failure.

Fatigue testing of the hygrothermally aged specimens was carried out with a cyclic loading range of 4.5–45% of the average batch tensile strengths of the aged specimen at 8 Hz. For virgin specimens, the cyclic loading range is based on the corresponding average virgin batch strengths. Again, the tests were interrupted periodically to allow the FBG spectra to be measured at 0 N.

## 3. Results

### 3.1. Spectrum Changes Due to Curing of an Adhesive Joint

On embedment into an adhesive joint, the FBG spectra normally exhibit some noticeable changes after the adhesive cured. Figure 3 presents two typical types of changes in the FBG spectra observed after embedment (solid lines) spectra versus those before (broken lines). The first type (Figure 3a) shows a slight shift of the spectrum and an emergence of secondary peaks (as pointed out by arrows). The second type is the occurrence of peak splitting and spectrum broadening as is evident in Figure 3b. The two spectra are from two different specimens. However, it is not uncommon to see both types from different FBGs from the same specimen. As explained before, the above changes in the FBG spectra are caused by the existence of non-uniform stress/strain on the FBG and are commonly observed in similar works [21,43,72,75].

During the curing process, the polymeric adhesive progressively transforms from a viscous liquid state to a hardened solid phase. This transformation is a chemical reaction that involves temperature change and volumetric contraction. The rate of cure, or chemical reaction, depends on a number of factors such as temperature, humidity and the availability of the correct stoichiometric ratio [78,79]. Local variation of these factors invariably exists, causing differential curing. Suppose a region A cured faster than its surrounding B. When A solidifies, its volume will contract and attain a certain dimension. Its shrinkage is not constrained by B as the latter is still in the viscous liquid phase and is relatively free to deform. However, when B solidifies or cures, its shrinkage will be constrained by A, which is not freely deformable. A residual compressive stress will result in A, while B will be subjected to a residual tensile stress. Differential curing rates at different distances from the FBG and at different locations along the FBG will therefore lead to a varying residual stress distribution acting on the grating. As the status of local variations differs, the pattern of residual stress distribution on each FBG will be different, leading to different patterns and different degrees of changes in the reflected spectra even among different FBGs in the same specimen. Similar changes in FBG spectra are also observed in other works [21,43,72,75] and FBG to FBG variations in different specimens are not uncommon [43,75].

### 3.2. Spectrum Changes during Hygrothermal Treatment

In the following section, the changes in the FBG spectra during the heating, soaking, cooling and drying stages are presented. Specimens A and B, respectively, denotes the same embedded FBGs as presented above in Figure 3a,b.

#### 3.2.1. Spectrum Change during the Heating Up Stage

Figure 4 shows the evolution of the two spectra when the specimens were heated up in the hygrothermal treatment water bath. As the water temperature increased from room temperature (RT) to 60 °C, the spectra showed a general shift towards the long wavelength direction. This is expected as explained in Section 2.1 above. In specimen A, the appearance of new emerging peaks and subsidence of old ones can be seen in Figure 4a. In Figure 4b, the split peaks, which appeared after embedment, merged into one when the temperature is raised to 40 °C. Temperature rose by about 1 °C/min and so the effect caused by infusion of moisture can at most account for a small part of the above phenomena. A more important effect may be attributed to thermal expansion. The latter to some extent offset the curing shrinkage and caused a redistribution of residual stress, which in turn led to the change in the FBG spectra.

#### 3.2.2. Spectrum Evolution during the Soaking Stage

Soaking at 60 °C in water lasted for 30 days. During this time, there were slight shifts and small changes of shape in the FBG spectra (Figure 5). Soaking in water encouraged moisture to diffuse into the polymeric adhesive and this should cause an expansion in volume and degrade the strength of the adhesive. The elevated temperature of 60 °C will both enhance the diffusion of moisture and cause the thermal aging of the epoxy adhesive. The latter tends to strengthen the adhesive and cause a contraction in volume [14,21,25,26]. The antagonistic effects of moisture-induced expansion and thermal aging-induced contraction probably resulted in a somewhat balanced effect on the volume and so the residual stress distributions are minimally affected. This may explain the insignificant changes observed in the FBG spectra in Figure 5.

#### 3.2.3. Spectrum Evolution during the Cooling down Stage

After soaking for 30 days at 60 °C, heating was stopped, and the specimens were allowed to cool down to room temperature before taken out from the water bath. Figure 6 shows some spectra from the two FBGs during this period. The room temperature spectra were measured in ambient air while the 40 °C and 60 °C spectra were measured in the water bath. In both cases, a general shift of the spectra towards the shorter wavelengths occurred. Following the explanation in Section 2.1, a 20 °C drops on a freely standing FBG will lead to a spectrum shift and peak wavelength decrease of ~200 pm. When temperature drops from 60 °C to 40 °C, Figure 6a showed a 320 pm shift and Figure 6b showed a 280 pm shift in the peak wavelength. The higher-than-expected shift may be caused by the thermal contraction of the adhesive which contributes to the additional shift towards the shorter wavelengths. Additionally, a change in positions of the emerging peaks can be observed in both cases as temperature decreases. The peak of the spectrum for specimen A in fact split into two on cooling to room temperature. These changes in the spectral shapes may be attributed to change in residual stress distribution associated with the thermal deformation.

Figure 7a,b compare the room temperature spectra of the two FBGs before and just after soaking. The spectrum from specimen A shows a marked broadening and the initial single-peak split into two distinct peaks after soaking (Figure 7a). That from specimen B goes the opposite way, showing a spectrum narrowing and that the initial split peaks combine into a single peak. These changes in the spectral shapes are probably affected by the following factors: (1) the initial residual distribution; (2) expansion of the polymeric adhesive due to absorption of moisture; (3) contraction of the polymeric adhesive due to thermal aging. (4) degradation that alters the stress–strain behavior and mechanical properties of the polymeric adhesive due to moisture and thermal aging. The spectra of both specimens after soaking tends to move toward the shorter wavelengths. This suggests that the contraction due to thermal aging seems to dominate over the expansion by moisture absorption in the current adhesive-adherend configuration.

The above tests have been repeated on 11 specimens, out of the 31 embedded optical fibers that survived, the spectra after soaking showed a general and often marked shift towards the shorter wavelengths as exemplified in Figure 7 in 28 FBGs. In the remaining three unbroken FBGs, one lost the reflection peak and two showed a slight shift towards the longer wavelength. The shapes of the spectra, both before and after soaking, were quite different among different FBGs. The initial differences in the shapes of FBG spectra may be attributed to a difference in residual stress distributions, which may result from local variations in curing of the adhesive that led to differential and mismatched contraction. On soaking, the moisture-induced expansion and thermal aging-induced contraction will interact with the initial mismatched contraction, leading to different outcomes from different initial conditions. This is probably the reason for different spectral shapes for different FBGs after soaking.

#### 3.2.4. Spectrum Evolution during the Drying Stage

After taken out from the water bath, wiped dry and had the as-soaked room temperature spectra taken, the specimens were prepared for tensile testing. Specimen A stayed in ambient air for about an hour during the above preparatory work. When the FBG spectra was again recorded before tensile testing commenced, it was found that prominent changes had occurred within the elapsed hour. The long wavelength edge of the spectrum had stayed the same but the short wavelength end had become heavily chirped and extended further towards the shorter wavelengths (Figure 8a). Prompted by this phenomenon, Specimen B was deliberately dried for a prolonged duration of 10 days in a desiccator. Figure 8b shows the evolution of FBG spectra during this period. A general shift and a progressively heavy chirping towards the shorter wavelength end occurred. The degree of change was large in the first four days. Beyond day 4, the growth of chirping on the short wavelength side has basically stopped and the width of the spectra stayed more or less the same. There were still variations in the intensities of different peaks within the chirped spectra with time. The nature of changes beyond day 4 may be illustrated by the day 5 and day 10 spectra shown in Figure 8b.

We have pointed out four effects on the FBG spectra after soaking. During the subsequent drying, the absorbed moisture gradually diffuses out of the adhesive and will offset the expansion effect. This may explain the general shift of the spectra to the shorter wavelengths. Thermal aging involves a series of change on the molecular level and the one that will occur first is post-curing [81]. Similar to initial curing, post-curing and other molecular changes will likely be non-homogeneous as well. This may further aggravate the non-uniformity of the residual stress. Additionally, the tendency for reduction in free volume and contraction associated with thermal aging [14,21,81] and moisture removal will reinforce the compressive side of the residual stress distribution. These two effects may explain the observed chirping and broadening of the spectra towards the shorter wavelengths.

From the above, we can see that the spectra from the embedded FBG are highly sensitive in revealing the internal changes caused by hygrothermal aging. The heavily non-uniform stress distributions make the conventional strain retrieval based on the peak wavelength shift impracticable. Interrogation technique such as that used in Reference [21] may provide more quantitative information. Alternatively, independent distributed fiber sensors, such as that made use of stimulated Brillouin scattering, may be used alongside the FBG to provide the strain distribution. For this purpose, their spatial resolution should preferrably be in the sub-millimeter level to yield meaningful results.

### 3.3. Damage Monitoring during Tensile Tests

#### 3.3.1. Spectrum Evolution in Virgin Specimens under Tensile Loading

In a previous paper [82], we have shown that tensile loading on virgin specimens progressively shifted the FBG spectra towards longer wavelengths without changing the spectral shape initially. As loading increased, some secondary peaks emerged and were enhanced with further increase in loading. When loading was above ~65% of the tensile strength, the spectra have markedly broadened or chirped. Sometimes the background intensity also rose significantly. Figure 9a shows a typical example that illustrate the above events. Broadening and chirping of the spectra is caused by a development of non-uniform strain distribution along the FBGs. It has also been pointed out that stress concentration occurs in the lap joint [83]. Both the stress concentration near the edges of the joint and the development of internal damage can give rise to non-uniform strains. Additionally, these two effects will be aggravated by increasing load, giving rise to the observed evolution of the spectra in Figure 9a. It is difficult to differentiate the contribution of loading alone and contribution of the damages in spectra measured at the instantaneous loading.

Reference [82] suggested that for damage monitoring, the effect of loading should be precluded and the spectra to be measured at the load-free condition. Occurrence of damages perturbs the residual stress field, leading to change in spectral responses. Figure 9b shows the unload spectra measured at 0 N from the same FBG after loading up and unloading from various tensile loads. Up to a loading of 65% of the strength, the unload spectra virtually overlapped with the reference spectra recorded initially at 0 N at the beginning of the tensile test, suggesting negligible damage has arisen at this stage. The 77% strength unload spectra exhibited slightly more enhanced secondary peaks when compared with the reference, indicating damage has probably commenced. The 97% strength unload spectra were heavily chirped, showing that damage has probably become extensive.

#### 3.3.2. Spectrum Evolution in Hygrothermally Damaged Specimens under Tensile Loading

After soaking at 60 °C for 30 days, the specimens were tensile tested after being left in ambient air (Specimen A) for one hour or dried for 10 days in a desiccator (Specimen B). The average virgin specimen batch strengths for specimens A and B are, respectively, 8.26 kN and 9.64 kN. Specimen A failed at 6.3 kN and specimen B at 5.8 kN. Figure 10a,b shows the respective evolution of spectra with increasing tensile load. For clarity, only some typical spectra are presented. Each loading is indicated by two numbers: as a percentage of the current specimen strength and of the average virgin specimen batch strength. The latter percentage is bracketed. For Specimen A, as loading was increased to 32% of current specimen strength, the initially markedly split peaks merged together to form a near single-peak spectrum with a number of emerging peaks on both edges. On further loading to 79% of current strength, the spectra broadened and chirped into multiple peaks. Besides the above changes in shape, the whole spectrum moves toward the longer wavelength as loading was increased. For Specimen B, the initial spectrum before test was already heavily chirped with multiple peaks. On loading up, the spectrum moved towards the longer wavelength while maintained the heavy chirping. The wavelengths and intensities of different peaks were changing with load.

When we view the loading as a percentage of the strength of Specimen A, the 32% and 79% strength spectra in Figure 10a are, respectively, similar to the 35% and 77% strength spectra of virgin specimen in Figure 9a. However, if the loading is viewed as a percentage of the average virgin batch strength (bracketed numbers in Figure 10), the heavy chirping of spectrum at 61% of virgin strength is much more serious than the corresponding 65% strength spectrum in the virgin specimen. In Specimen B, the heavily chirped and broadened spectra under all loading are vastly different from that in virgin specimens. These are understandable as hygrothermal treatment probably brought about considerable damage and weakened regions, leading to premature appearance of heavy chirping of the spectra.

In Figure 11, we preclude the effect of loading and look at the effect of tensile damage through the unload spectra. The loading reached before unload was again indicated as a percentage of the current specimen strength and of the average virgin specimen batch strength. For Specimen A, from ~12% of strength, the chirped pattern of the initial spectrum before testing started to change with load but the width of the spectra remained more or less the same. Peaks near the short wavelength end decrease, while that on the long wavelength end gain in intensity. On reaching 32% of current specimen strength and unload, a slanted peak resulted (Figure 11a). Further loading to 79% strength did not change the short wavelength side of the unload spectra significantly but broadened it a little bit on the long wavelength side. On loading further, the spectra again chirped into a number of peaks before final failure.

Figure 11b compares three instances of the unload spectra with the reference spectrum. Throughout the tensile test, the width of the spectra remained basically the same, but the intensities of different peaks are changing continuously with different loads.

The change in intensities of the peaks in both Figure 11a,b were caused by a variation in the stress distributions on the FBGs. The shape of the initial spectra reflects the initial residual stress distributions after hygrothermal treatment. Tensile loading was most likely caused by damages at locations with a combination of high local stress and weakened strength. These damages did not occur uniformly, and the residual stress distributions were continuously perturbed and redistributed under different tensile loads. Remnant of such perturbations persist even though loading is removed, leading to variations in the intensities of different wavelengths of the chirped spectra.

In the above tests, the strengths of the specimens were eventually known as the specimens failed. This is not the case in real life adhesive bond integrity monitoring applications as the strengths of environmentally damaged in-situ structures are difficult if not impossible to know in advance. On the other hand, the virgin structural strengths can normally be obtained beforehand through in-house testing. Thus, in judging the spectral responses under load, it will be more meaningful to view it in terms of a percentage of the virgin strength rather than the current specimen strength. When compared with the virgin specimen in terms of a percentage of the average virgin batch strength, the spectral responses of the hygrothermally damaged specimens exhibit a number of phenomena: (1) before loading was applied, a marked shift and change in the chirp status usually occurred with regard to the as-embedded spectra; (2) under load, heavy chirping of the spectra started to occur at a smaller percentage of virgin strength; (3) the unload spectra show marked change in the shape on small loading (~10–~35% of virgin strength), while similar change occurred at >~75% strength in virgin specimens. If integrity degradation of the adhesive bond by high temperature and humidity is suspected, the above phenomena may act as some diagnostic indications.

### 3.4. Damage Monitoring during Cyclic Fatigue Loading

#### 3.4.1. Virgin Specimens Fatigue Testing

The average fatigue lives of three different batches of virgin specimens tested under a cyclic load range between 4.5% and 45% of their average batch failure strengths is 184,200 cycles, with a standard deviation of 43,273 cycles. Figure 12 shows two typical unload spectra evolution during the course of fatigue life reported in reference [82]. These are from FBGs embedded on the left side (FBGL) and in the centerline (FBGM) from the same specimen. For FBGM, the change in the spectra was very gradual, up to 120,000 cycles, or ~71.5% of life. There was only a slight shift in peak wavelength and appearance of weakly discernible secondary peaks. This kind of development was maintained until the last spectra were taken at 165,000 cycles, which is already ~98.2% of fatigue life. For FBGL, a gradual rise in the background intensity occurred. This rise can be observed starting from 5000 cycles. However, apart from the raised background intensity, the remaining part of the spectrum stayed in more or less the same shape up to 75,000 cycles, or ~45% of life. At 120,000 cycles, slight changes in the emerging peaks on this remaining part was observed. Rising background intensity indicates damages were probably occurring but the small change in the other part of the spectra suggests damage was not serious yet.

In comparison with tensile damage, the expression of damage in the FBG spectra is much gentler under fatigue. In tensile failure, the large loading coupled with the stress concentration effects of the joint and the defects will cause widespread damage, giving rise to very marked change in the FBG spectra considerably before the final failure. On the other hand, fatigue damage is highly localized and stochastic in nature. Under small cyclic loading, initial damages develop locally and slowly. If such damages happen to not be occurring right in the close vicinity of an FBG, the change in the spectra will not be marked. Extensive spread of damage to all over the joint may probably occur extremely close to the end of life. Periodic recording at an increment of a few thousand cycles may easily miss the final stages of failure. This explains the relatively gentle changes in the spectral responses observed in Figure 12.

#### 3.4.2. Fatigue Testing of Hygrothermally Aged Specimens

Similar to the tensile specimens A and B, the fatigue specimens C and D were soaked in 60 °C water for 30 days and Figure 13a,b compare the room temperature spectra of the two FBGs before and just after soaking. The spectrum of specimen D showed a marked shift towards the shorter wavelengths, broadening with multiple peaks and a decrease in intensities of the peaks. Specimen C remained a single peak with a slight shift to the shorter wavelengths. The difference in spectral changes are the combined results of a number of factors as discussed in Section 3.2.3.

Specimen C was tested within one hour, during which its spectrum continued to show a very slight shift towards the left (Figure 14a). Specimen D was dried in a desiccator for 10 days. Figure 14b shows the spectrum evolution during this drying period. It continued its broadening, chirping, shifting to the left and deceasing in peak intensity. By the fifth day, shifting and broadening had nearly stopped but the drop in intensity continued. This trend of development and the resulting heavily broadened spectrum with multiple peaks is quite similar to that of Specimen B, which was treated in the same way.

Specimens C and D were subsequently subject to cyclic loading. Rather than using the virgin batch strength to define the cyclic loading, they were tested with a cyclic load range between 4.5% and 45% of the average strength of the same batch of specimens after soaking. For Specimen C, the unload spectrum after 1000 cycles showed a significant increase in intensity at the long wavelength end (Figure 15). This increase continued but at a slower rate up to 20,000 cycles. By 25,000 cycles, a marked rise in the background intensity occurred. The shape of the remaining part of the peak also deviated from the initial shape. These two phenomena continued to develop as loading cycles increased. The last spectrum recorded was at 37,000 cycles. The specimen failed shortly after at 37,380 cycles. Four hygrothermally treated specimens have been tested in the same way as Specimen C and their average fatigue life was 37,107 cycles with a standard deviation of 16,046 cycles. This was considerably shorter than the 184,200 cycles of the virgin specimens, even though the cyclic load ranges were chosen to be the same percentage of their respective average batch strengths. For Specimen D, we aimed to record the first unload spectrum at 500 cycles, but it failed after 320 cycles.

Although the tensile strength and fatigue life of the desiccator dried specimens happened to be respectively lower than that of the as-soaked specimens, the number of dried specimens tested was too limited to draw a solid conclusion on whether they really have inferior tensile and fatigue strength to the as-soaked specimens. However, from the heavily broadened multi-peaked spectra of the FBGs, it may be concluded that the residual stress distributions are much more seriously perturbed, indicating that the weakening and damages were more extensive in the dried specimens than in the as-soaked ones.

## 4. Conclusions

The development of the full spectral responses of uniform period FBG sensors embedded in epoxy adhesive single-lap joints during hygrothermal aging at 60 ℃ for 30 days of the joint specimens and their subsequent tensile and fatigue failures has been studied. The key findings may be summarized as follows:(1)Shifting, emergence of secondary peaks and broadening/chirping of an initially single narrow-peak spectrum occurred after embedment, presumably due to cure-induced, non-uniform residual stress.(2)When compared under room temperature, the current hygrothermal aging caused a marked shift toward the shorter wavelengths and change in the chirp status with respect to the as-embedded spectra. Subsequent drying led to heavy chirping and broadening towards the short wavelength end.(3)Under tensile load, the aged specimens have lower strengths. Marked change in the shape of the unload spectra already took place at ~10–~35% of virgin strength and heavy chirping of the spectrum was already evident at ~35% of virgin strength. The corresponding changes in virgin specimens occurred at greater than ~75% strength.(4)Under cyclic loading, the hygrothermally aged specimens have markedly shorter fatigue lives and marked broadening and deviation in the shape of the unload spectra was evident at ~3% of life or 1000 cycles. The corresponding changes in virgin specimens occurred at greater than ~70% life.

When a composite structure with adhesive joints is in a susceptible environment, evidence of hygrothermal degradation may be revealed by a change in the chirp status and a shift of the spectrum away from the as-embedded spectrum of an FBG. Appearance of heavy chirping in the unload spectra after loading to a small fraction of virgin strength or after cyclic loading for a small fraction of fatigue life also signifies the occurrence of hygrothermal degradation.

## Figures and Tables

**Figure 1 polymers-14-00368-f001:**
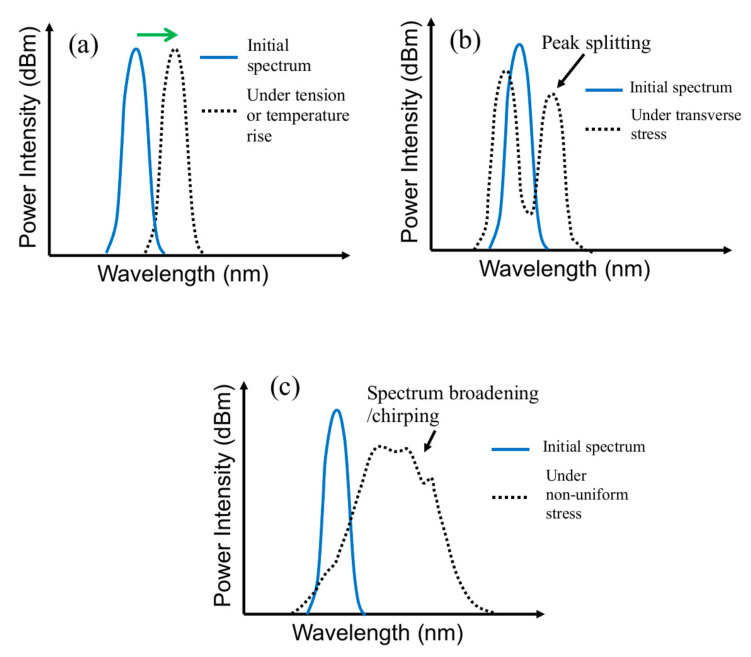
Schematic diagram to show FBG spectrum changes under (**a**) tensile stress or temperature rise; (**b**) transverse stress and (**c**) non-uniform stress along its length.

**Figure 2 polymers-14-00368-f002:**
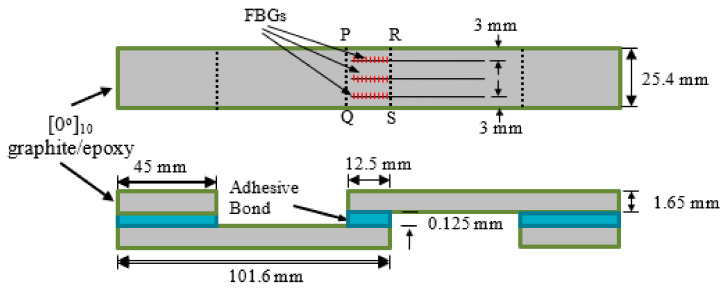
Dimensions and layout of the single-lap joint specimen.

**Figure 3 polymers-14-00368-f003:**
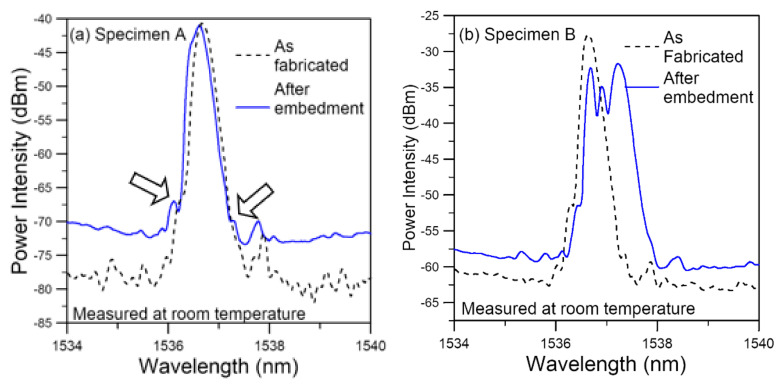
Typical FBG spectrum changes after joint curing: (**a**) a slight shift and an emergence of secondary peaks (pointed out by arrows); (**b**) peak splitting and spectrum broadening.

**Figure 4 polymers-14-00368-f004:**
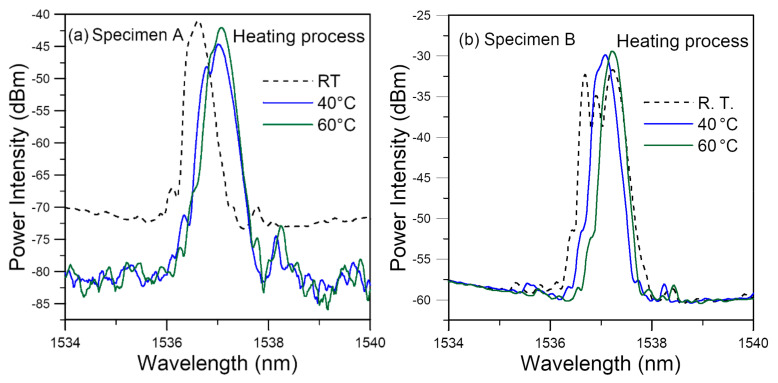
Embedded FBG spectra evolution during the heating process before soaking showing general shift to the right and (**a**) changing of emerging peaks and (**b**) merging of split peaks.

**Figure 5 polymers-14-00368-f005:**
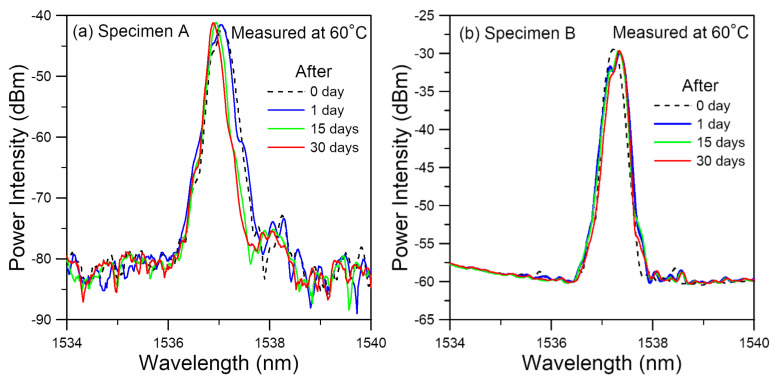
FBG spectra shows little changes during the soaking process in both specimens (**a**) A and (**b**) B, probably due to the mutual balance between the antagonistic effects of moisture-induced expansion and thermal aging-induced contraction.

**Figure 6 polymers-14-00368-f006:**
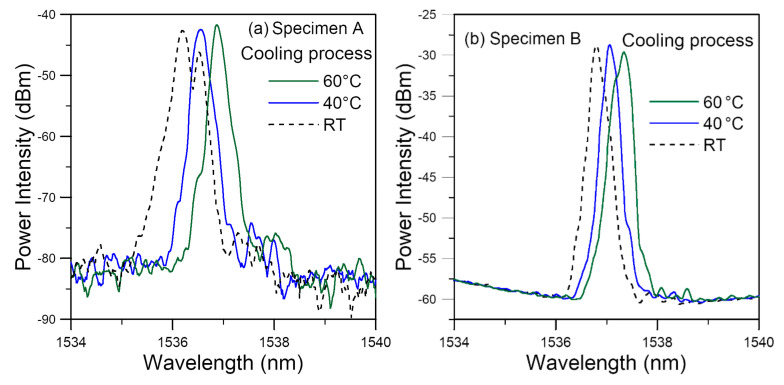
FBG spectra during the cooling process after soaking at 60 °C for 30 days showing general shift to the left for specimens (**a**) A and (**b**) B.

**Figure 7 polymers-14-00368-f007:**
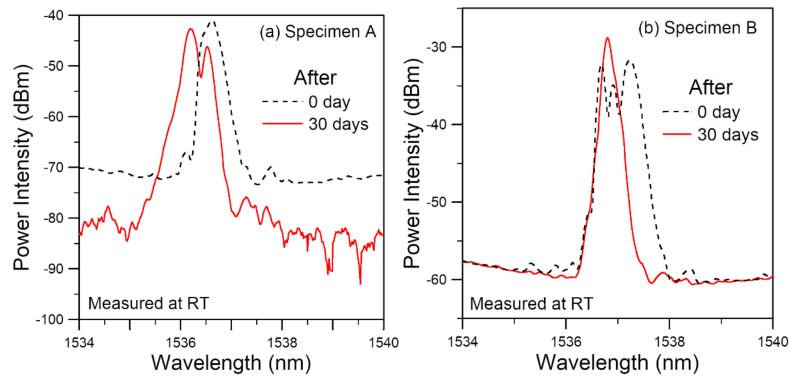
Comparison of room temperature FBG spectra before and just after soaking at 60 °C for 30 days from specimens (**a**) A, which shows a marked broadening and peak splitting; (**b**) B, whichs shows a spectrum narrowing and merging of split peaks.

**Figure 8 polymers-14-00368-f008:**
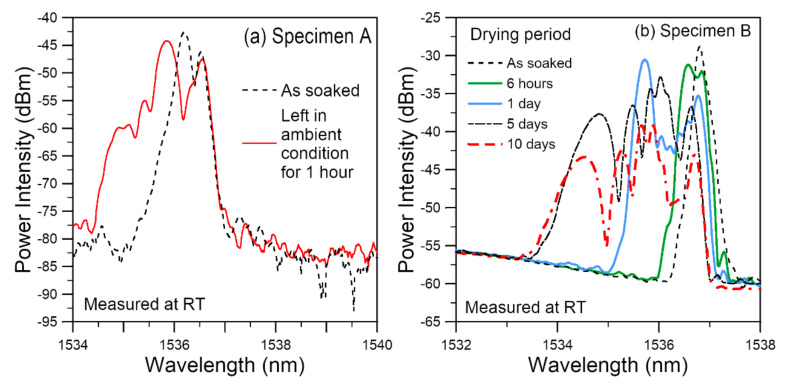
Evolution of FBG spectra during drying in: (**a**) ambient condition for 1 h; (**b**) in a desiccator for 10 days.

**Figure 9 polymers-14-00368-f009:**
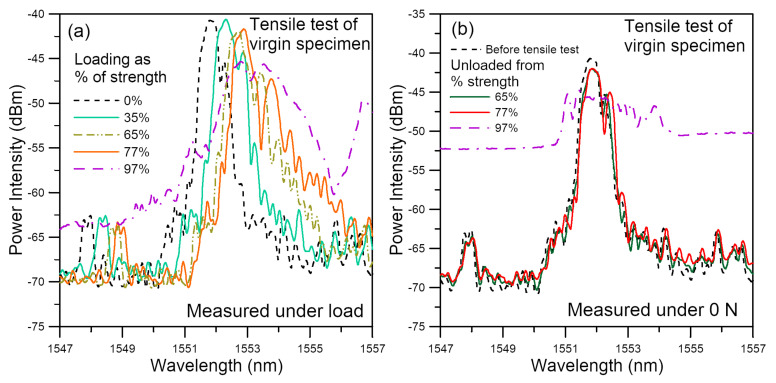
Evolution of virgin specimen spectra under tension measured (**a**) at various loads; (**b**) at 0 N after unloading from various tensile loads.

**Figure 10 polymers-14-00368-f010:**
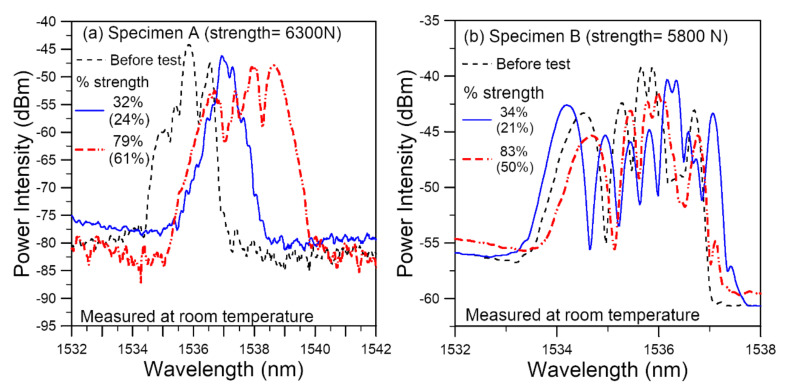
FBG spectra of hygrothermally treated specimens under various tensile loads shown as a percentage of the current specimen strength as well as percentage of the average virgin specimen batch strength (bracketed numbers) from (**a**) specimen A and (**b**) specimen B.

**Figure 11 polymers-14-00368-f011:**
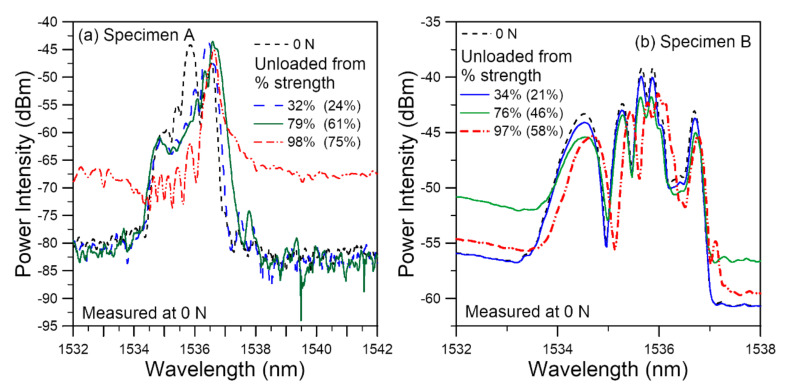
FBG spectra of hygrothermally treated specimens measured at 0 N after unloading from various tensile loads shown as a percentage of the current specimen strength as well as a percentage of the average virgin batch strength (bracketed numbers) from (**a**) specimen A and (**b**) specimen B.

**Figure 12 polymers-14-00368-f012:**
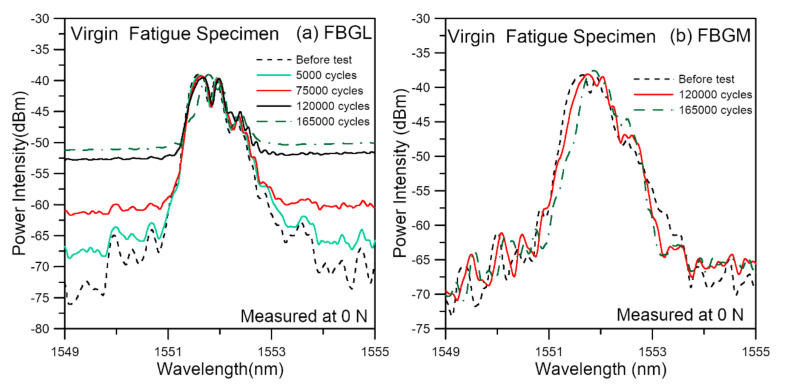
Evolution of typical unload FBG spectra from (**a**) FBGL and (**b**) FBGM, during fatigue cycling of a virgin specimen.

**Figure 13 polymers-14-00368-f013:**
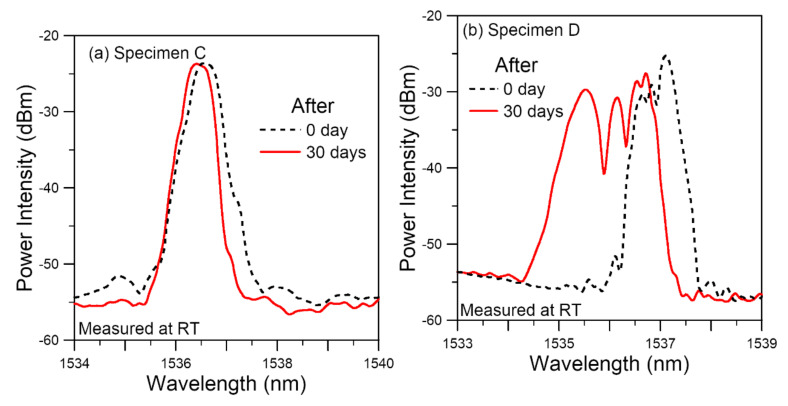
Comparison of room temperature FBG spectra before and after soaking at 60 °C for 30 days from Specimens (**a**) C, which shows a slight spectrum narrowing; (**b**) D, which shows a marked broadening and peak splitting.

**Figure 14 polymers-14-00368-f014:**
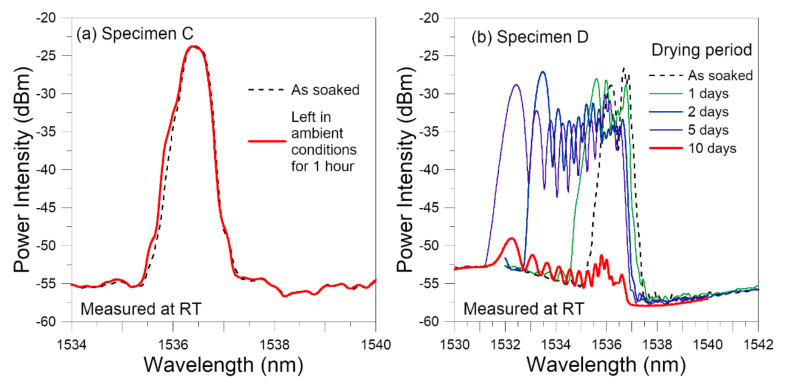
Evolution of FBG spectra from the two fatigue specimens during drying in: (**a**) ambient condition for 1 h; (**b**) in a desiccator for 10 days.

**Figure 15 polymers-14-00368-f015:**
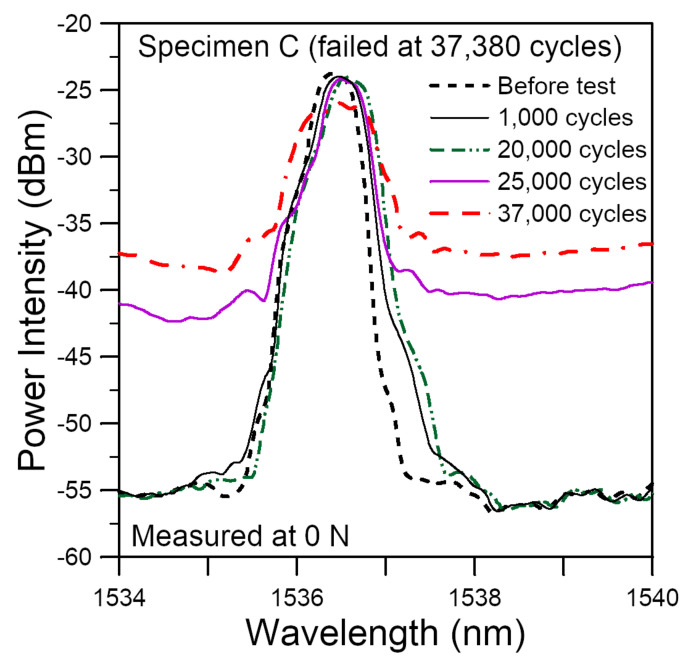
Evolution of typical unload FBG spectra during fatigue cycling of a specimen soaked at 60 °C for 30 days.

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
