# Peer review of "Hygrothermal Damage Monitoring of Composite Adhesive Joint Using the Full Spectral Response of Fiber Bragg Grating Sensors"

_polymers, 2022, doi:10.3390/polym14030368_

Round 1
Reviewer 1 Report
Review article:
Hygrothermal damage monitoring of composite adhesive joint using the full spectral response of fiber Bragg grating sensors
The authors propose an humidity Hygrothermal damage monitoring of composite adhesive joint using the full spectral response of fiber Bragg grating sensors.
This is a well written and structured manuscript. In this manuscript there are the potentialities to publish on Polymers but the document needs some specifics and reviews.
At line 136 n is the refractive index of core (you need to adding this information)
The behavior of the FBG sensor is very different when inserted in a packging. So there is a need to know the behavior before and after the insertion in the packging.
The Ylabel of Figure5, Figure 8, Figure 9, Figure 10, Figure 11, Figure12, Figure 15.
To better characterize the structure with FBGs, it would be very useful to use an observable such as the conventional observable as Bragg wavelength. In this way it will be possible to obtain the Bragg wavelength as the strength or cycle or temperature or specimen varies.
Author Response
Hygrothermal damage monitoring of composite adhesive joint using the full spectral response of fiber Bragg grating sensors
The authors propose an humidity Hygrothermal damage monitoring of composite adhesive joint using the full spectral response of fiber Bragg grating sensors.
This is a well written and structured manuscript. In this manuscript there are the potentialities to publish on Polymers but the document needs some specifics and reviews.
- At line 136 n is the refractive index of core (you need to adding this information)
Thank you for point this out. It has been corrected and the sentence (line 150) becomes “where n is the effective refractive index of the fiber core.”
- The behavior of the FBG sensor is very different when inserted in a packging. So there is a need to know the behavior before and after the insertion in the packging.
It is believed that the FBG after insertion still behaves according to equation 1. For this reason Refs. 43, 44, 72 and 73 used the embedded FBG wavelength shift to monitor the hygrothermal swelling strain and Refs. 60-66 used it to monitor the strain under mechanical loading. In our own test on embedded FBG, on application of a series of loading, small enough not to cause internal damage, the spectrum shifted according to equation 1 but preserved the same shape. This supports the view that the fundamental characteristics of FBG are not changed by insertion into composites.
It should be noted that shape of the spectrum does change after embedment, as the FBG is then subject to residual stresses caused by the curing. Subsequent internal damages further change the spectrum shape due to occurrence of stress redistribution and this have been made use of to detect damages in the current work. We have actually tried to remove the internal stresses on the FBG by soaking fatigue damaged joint specimens in acetone. The pre-embedment spectra were restored after soaking for a few day when the adhesive was softened. We have also tried to extract the FBG from damaged composite specimen. Not only did the spectrum restored but also on straining the spectrum shifted according to equation 1. The latter results has been reported in “Shin, C.-S.; Liaw, S.-K.; Yang, S.-W. Post-Impact Fatigue Damage Monitoring Using Fiber Bragg Grating Sensors. Sensors 2014, 14, 4144-4153.”
- The Ylabel of Figure5, Figure 8, Figure 9, Figure 10, Figure 11, Figure12, Figure 15.
Thank you for pointing this out. We are not aware that the auto-generated line numbers interferes with the y-labels. The above figures have now been re-positioned.
- To better characterize the structure with FBGs, it would be very useful to use an observable such as the conventional observable as Bragg wavelength. In this way it will be possible to obtain the Bragg wavelength as the strength or cycle or temperature or specimen varies.
We totally agree that it will be very useful to identify a unique parameter from the FBG spectrum to correlate with strength, fatigue life and so on. We have tried in vain to find one under hygrothermal aging. In purely mechanical loading cases, we have identified a V value that quantifies the amount of change in spectral shape and correlates with the degree of damage. This result has been reported in a companion paper (Ref. 82). Under hygrothermal aging, this parameter cannot give consistent prediction as heavy chirping brought drastic and highly irregular changes in the shape of the spectra. The latter phenomenon also precluded a representative wavelength to be identified to quantify the degraded mechanical properties.

Reviewer 2 Report
This paper makes a stoichiometric analysis of chemical or mechanical features of composite material by means of a relatively long FBG. The analysis conducted is quite qualitative but may be informative for material science people though little for FBG or optical people.
Author Response
This paper makes a stoichiometric analysis of chemical or mechanical features of composite material by means of a relatively long FBG. The analysis conducted is quite qualitative but may be informative for material science people though little for FBG or optical people.
Thank you very much for the comment and we totally agree with it. In fact, the paper is aimed at material scientists/mechanical engineers in search of a way to monitor the structural health degradation under hygrothermal aging.

Reviewer 3 Report
This work reports a method for monitoring composite adhesive joints using fiber Bragg grating (FBG) sensors. The authors conducted several tests to correlate the fiber spectrum changes with the stress distribution along with the sample. Albeit the manuscript is well-written and provides valuable information to the journal’s audience, there are points worth noticing:
- Sec. 1: Please emphasize the novelty and scientific advances of the proposed methodology beyond the state-of-the-art, especially regarding the optical fiber sensors mentioned in the last paragraph of the Introduction;
- Fig. 2: Remove the underlines below units in the x and y labels (for example, nm);
- Sec. 2.1: Include a reference to explain the FBG fabrication method;
- Sec. 2.2: Did you embed standard telecom fibers inside the composite structure? Or did you attach the fibers onto the laminate surface? What is the period (L) of each FBG?
- Sec. 3: The results for testing the samples to several temperature, humidity, and strain cycles/conditions demonstrate the feasibility of using FBG sensors to dynamically monitor the composite structure behavior. However, it is hard to retrieve quantitative information from the output spectra due to the non-uniform loading. Is it possible to obtain the stress distribution (in Pa) along with the sample by combining the response of the three FBGs? For instance, one may attempt to conduct such a characterization with a stimulated Brillouin scattering-based distributed sensor;
- Sec. 3: The FBG spectra contain superposed mechanical and thermal effects. Consider using another FBG to correct the temperature shift and measure the pure stress distribution;
- Sec. 3: I advise the authors to conduct comparative mechanical tests and microscopy analysis to validate the FBG results and strengthen your discussions.
Author Response
This work reports a method for monitoring composite adhesive joints using fiber Bragg grating (FBG) sensors. The authors conducted several tests to correlate the fiber spectrum changes with the stress distribution along with the sample. Albeit the manuscript is well-written and provides valuable information to the journal’s audience, there are points worth noticing:
- 1: Please emphasize the novelty and scientific advances of the proposed methodology beyond the state-of-the-art, especially regarding the optical fiber sensors mentioned in the last paragraph of the Introduction;
In order to emphasize the novelty and scientific advances of the proposed methodology beyond the state-of-the-art as advised, some relevant descriptions scatter among the previous passages have been pulled together and reinforced to become a new paragraph. It is added before the last paragraph of the Introduction. The new paragraph added is:
“The use of FBGs for hygrothermal aging monitoring has a number of distinct advantages over conventional sensors. These include a much better compatibility with the host materials and will not behave as defects when embedded, the ease of multiplexing, as well as immunity from electromagnetic interference and environmental attack. Moreover, the grating period that responds to the surrounding strain is in the order of micrometer. When embedded its close proximity to the degraded materials/defects together with its small responding gage lengths make FBGs very suitable for reflecting small local perturbation of strain caused by material changes and degradation. Instead of the normally employed single peak wavelength that only indicates the swelling strain, it is postulated the full spectral response of FBGs will be much more capable in revealing the onset of hygrothermal degradation as well as the development of subsequent aging damages. The clearer picture about the degradation status so obtained can help to avoid uneconomical premature retirement of components and precarious use of heavily degraded structures. ”
- 2:Remove the underlines below units in the x and y labels (for example, nm);
Thank you for pointing this out. The underlines below units in the x and y labels in Fig.2 has now been removed.
- 2.1: Include a reference to explain the FBG fabrication method;
A reference 81 on the side writing technique to fabricate FBG has now been included. Line198-199 has now been modified as :”In this work, single peak FBGs were used and were fabricated in a Ge-B co-doped single mode optical fiber by side writing using a phase mask [81].” The number of the following references has been corrected accordingly.
- 2.2: Did you embed standard telecom fibers inside the composite structure? Or did you attach the fibers onto the laminate surface? What is the period (L) of each FBG?
The sensing fibers are embedded inside the adhesive joint and are of the same size as the standard telecom fibers, i.e. 125μm in diameter. This has been stated in lines 217-219. They are a special Ge-B co-doped photosensitive fibers, as stated in line 198. The grating period is 0.526μm.
- 3: The results for testing the samples to several temperature, humidity, and strain cycles/conditions demonstrate the feasibility of using FBG sensors to dynamically monitor the composite structure behavior. However, it is hard to retrieve quantitative information from the output spectra due to the non-uniform loading. Is it possible to obtain the stress distribution (in Pa) along with the sample by combining the response of the three FBGs? For instance, one may attempt to conduct such a characterization with a stimulated Brillouin scattering-based distributed sensor;
The FBGs are extremely sensitive to local variations and so it is not practicable for evaluating the overall stress distribution.
Theoretically, it is possible to use stimulated Brillouin scattering-based distributed sensor alongside with the FBG to measure the local strain concurrently. The practical difficulty here is the spatial resolution of the distributed sensor. Literatures show that the best available resolution is in the millimeter range. Stress analysis [84] indicated that near the edge of adhesive joints a very steep stress gradient that calls for a much better spatial resolution to quantify it. In the vicinity of a crack-like disbond or defect, a very steep stress gradient is again expected. The heavily chirped spectrum exhibited by the FBG in fact reflected these situations. As a result a stimulated Brillouin scattering-based distributed sensor can help to give a general picture of the average stress but not the actual stress distribution. In fact stress distribution along the FBG may be obtained by using more advanced measurement technique such as that mentioned in [21]. Unfortunately such equipment is not currently available to us.
- 3: The FBG spectra contain superposed mechanical and thermal independent monitoring of the effect of temperature on the FBG spectrum. As effects. Consider using another FBG to correct the temperature shift and measure the pure stress distribution;
This is indeed the case in practical situation and it really calls for an independent monitoring of the effect of temperature on the FBG spectrum. As mentioned in lines 202-206, our experiment, except during hygrothermal treatment, the other tests were conducted in a air-conditioned room with temperature control to within ±1°C and the FBGs were embedded in poor thermal conductors of polymeric adhesive sandwiched between composite laminates so that it is relatively insensitive to outside temperature changes. Small ambient temperature fluctuations will have negligible effect on the measured spectra during mechanical testing in our case. However, obtaining the stress distribution calls for other more advanced measuring techniques not currently available to us.
- 3: I advise the authors to conduct comparative mechanical tests and microscopy analysis to validate the FBG results and strengthen your discussions.
Thank you very much for your advice and indeed this will be very helpful towards more thorough understanding of the current topic. We are currently trying to secure funding for a new project aiming at relating the course of development of the FBG spectra with micro-mechanisms happening inside the composite during hygrothermal aging. In view of the occurrence of different and competing mechanisms during aging, besides elevated temperature high humidity, control experiments at elevated temperature low humidity, room temperature high humidity as well as wet-dry cycling are needed to differentiate the different interacting mechanisms. At different stages of aging and damages under mechanical loading, the specimens will be sectioned open to examine the damage in a scanning electron microscope.

Round 2
Reviewer 1 Report
The authors filled in the gaps
Reviewer 3 Report
The authors improved the manuscript following the reviewers' suggestions. As a final observation, I suggest including the comments about distributed stress measurements and temperature sensitivity (queries 5 and 6 of reviewer 2) in the discussion or conclusion sections.